# Factors Affecting Avatar Customization Behavior in Virtual Environments

Sixue Wu [1,2,\*], Le Xu [2,3], Zhaoyang Dai [4] and Younghwan Pan [2,\*]

1   College of Chinese and ASEAN Arts, Chengdu University, Chengdu 610106, China
2   Department of Smart Experience Design, Kookmin University, Seoul 02707, Republic of Korea
3   College of Design, Zhijiang College of Zhejiang University of Technology, Shaoxing 312030, China
4   London College of Communication, University of the Arts London, London SE16SB, UK
\*   Correspondence: wusixue@kookmin.ac.kr (S.W.); peterpan@kookmin.ac.kr (Y.P.)

**Abstract:** This research aims to examine the psychology and behavior of users when customizing avatars from the standpoint of user experience and to provide constructive contributions to the Metaverse avatar customization platform. This study analyzed the factors that affect the behavior of user-customized avatars in different virtual environments and compared the differences in public self-consciousness, self-expression, and emotional expression among customized avatars in multiple virtual contexts. Methods: Using a between-subjects experimental design, two random groups of participants were asked to customize avatars for themselves in two contexts, a multiplayer online social game (MOSG) and a virtual meeting (VM). Results: When subjects perceived a more relaxed environment, the customized avatars had less self-similarity, and the subjects exhibited a stronger self-disclosure willingness and enhanced avatar wishful identification; nevertheless, public self-consciousness was not increased. When subjects perceived a more serious environment, the customized avatars exhibited a higher degree of self-similarity, and the subjects exhibited a greater self-presentation willingness, along with enhanced identification of avatar similarity and increased public self-consciousness. Conclusions: Participants in both experiment groups expressed positive emotions. The virtual context affects the self-similarity of user-customized avatars, and avatar self-similarity affects self-presentation and self-disclosure willingness, and these factors will affect the behavior of the user-customized avatar.

**Keywords:** virtual environment; avatar customization; avatar self-similarity; avatar identification; public self-consciousness; online self-expression; emotional expression





## 1. Introduction

Although virtual worlds have existed since the 1970s, they have grown increasingly prevalent during the last decade as a result of 3D modeling, rich visual design, and multimodal interaction capabilities. Popular types of virtual worlds for many years include MMORPGs, such as World of Warcraft and Second Life. During the COVID-19 pandemic, Roblox, a sandbox game based on user-generated content, enabled access to the metaverse. Whether it is a concert in Fortnite, a graduation ceremony in Minecraft, or an academic conference at the Animal Crossing Society, avatars serve as the entry point for humans into virtual environments. A common objective of avatar design in many virtual settings is to incorporate the real or intended traits of the user into the avatar, thus enhancing the user's overall perception of the environment and engagement with it [1]. Numerous digital media user interfaces now allow users to build and use personal avatars for participation in online environments. These avatars may be used for a multitude of reasons, including gaming, e-commerce, online education, and social networking. Users are afforded more latitude in the areas of self-expression and identity expression as a result of the ability of avatars to modify, manipulate, and personify their digital personas [2]. By clicking, touching, or dragging the customizable choices in the avatar creation interface, users may alter many

characteristics of their avatars, including bodily parts, facial features, clothes, and more, to produce new forms of online self-expression.

Avatar communication in virtual social environments can stay anonymous and accommodate a variety of nonverbal expressions [3]. Users can self-disclose in the virtual world to compensate for what they lack in the actual world, or they can show their genuine selves outside of social roles [4,5]. Numerous studies have demonstrated that allowing users to customize their avatars enhances player enjoyment, engagement, and presence [6–8]. Moreover, the attractiveness of avatars impacts user loyalty [2,9]. The vast majority of research on avatars is focused on the virtual environment of entertainment games; therefore, multiplayer online social games are one of the virtual settings examined for tests in this study. The COVID-19 epidemic has led to a significant surge in telecommuting and distance learning. As web conferencing gadgets seem to be a pervasive communication tool in socially distant lives, "zoom fatigue" tales have spread rapidly [10]. Major technology firms have started to build and grow online virtual meetings to increase the interactive experience of immersive virtual meetings. There is limited study on avatars in virtual conferences at now. A virtual meeting is another experimental virtual setting investigated in this research.

This study examines the processes involved in customizing avatars in various virtual environments. To explore and expand the potential value of avatars in identification, it is anticipated that customizing avatars in diverse virtual contexts will arouse public self-consciousness to varying degrees. Users frequently use self-expression and identity to convey their self-concept through avatars [11]. Personalization of avatars is thus a deliberate act of self-expression and definition. Self-similarity and avatar identification may vary between different virtual environments for constructed avatars, and the expression of emotions when customizing an avatar may vary depending on the context.

This study utilized a between-subjects experimental design: group 1 customized avatars for multiplayer online social games on the same platform as group 2 customized avatars for work-study virtual meetings. We compared the avatar self-similarity, the manipulation duration of each design element, the public self-consciousness and self-expression of the subjects during the avatar customization process, and the customized avatar identification in the two experimental groups.

These are the research questions on which we are focusing:

RQ1: When users use the same avatar customization platform to customize avatars for different virtual environments, will the avatar self-similarity be different?

RQ2: Does avatar self-similarity affect public self-consciousness and avatar identification?

RQ3: Do customized avatars express different emotions in different virtual contexts? Do design elements affect emotional expression?

The following are many unique contributions of this study: According to Trepte and Reinecke [12], avatar similarities can improve computer game user experience. This impact is contingent on the amount of competitiveness in the game, with players choosing distinct avatars in competitive games and similar avatars in noncompetitive games. This research offers a fresh viewpoint on avatar customization in the context of social gaming and online meetings. Making an avatar look human-like may be accomplished in a variety of ways. Examples include hair and apparel that complement one another [6]. However, the majority of experiments were conducted with 2D avatars, and the realistic or stylized appearance of the avatars impacted the study outcomes. This experiment employs three-dimensional experimental materials, and the design style of the avatar mixes realism and stylization. The disparities in avatar self-similarity in several virtual settings will pave the way for future studies on the design of avatars suitable to numerous contexts.

Second, people in virtual worlds are hesitant to divulge their identities if their avatars appear realistic [13,14]. Based on this perspective, this study examines whether participants' public self-consciousness differs between formal, no-nonsense virtual environments and casual, liberal virtual environments, which may influence whether participants use avatars to disclose or present themselves. The behavior of customizing an avatar affects its

recognition. Due to the varying identifiability of avatars, it may be necessary in the future to combine additional avatar-based characteristics as a requirement for user selection.

Third, Takano and Taka [15] found that changes in facial features, such as facial position, contour, shape, and eyebrows, negatively affect avatar recognition. These aspects are difficult to alter in the real world. In contrast, hair parts that frequently transform in the real world are intrinsic to each individual's identity. We compared the avatars created in the two contexts to determine if the conclusions of prior studies had changed. Numerous studies have confirmed that human emotions can be expressed through cartoon characters [16] or human-like avatars [17], and through the gestures and facial expressions of the avatars [18]. Negative emotions can be reduced, and emotional relief promoted during interactions with virtual digital humans [19]. In the discussion of related work, we focus primarily on observing the differences in emotional expression when customizing avatars for different contexts and on determining whether avatar design elements affect emotional expression. Emotional expression improves the recognition of avatars. For creators of avatar customization platforms, the significance of design elements, the time required to repeat operations, and the emotional expression of the user will be crucial factors to consider.

## 2. Theoretical Background and Hypotheses

### 2.1. Subsection Avatar in Virtual Social Environment

In prior research, scholars discovered that despite technical limitations, people prefer to be in control of their avatar design [20]; avatar customization can make digital gaming experiences more enjoyable [21]; and people actually spend a great deal of time customizing their avatars to represent identity-related characteristics when interacting with others online [22–24]. Researchers investigating this phenomenon have investigated how circumstances, present emotions, and the desire to impress might lead to the construction of different roles. In the virtual world, identities are more malleable and varied, and individuals have the possibility to have character experiences that are not attainable in real life [25]. Vasalou and Joinson [11] discovered that avatars on blogging sites correctly reflected the look, lifestyle, and tastes of their respective owners. Participants on dating and gaming platforms, on the other hand, highlighted other features of their avatars. Avatars used in games, for instance, tend to seem more powerful or intelligent, whilst those used in dating simulations tend to appear more appealing. This shows users' avatar choices rely on communication objectives and virtual environments [11,26,27]. Previous research has shown that the correct use of avatars in online video conferencing, together with self-affirmation, has a positive impact on active participation in discussions [28].

### 2.2. Subsection Avatar Customization and Similarity

Avatars are often defined as the interactive mediators between users and self-visual descriptions in virtual environments [29]. Thus, avatars enable users to experiment with different identities in a virtual environment [30]. Avatars may be pre-programmed stock pictures by professional developers or unique representations made by the users themselves utilizing in-built art tools [31]. In recent years, a lot of progress has been made in terms of how much an avatar may be customized. Many virtual worlds enable members to utilize AI to produce avatars by capturing images. Users may adjust the avatar's skin color, eye color, haircut, height, body type, clothes, accessories, and personality qualities. Users have the ability to create a self-image that is distinct in appearance and can be customized via the usage of these elements, which facilitates their participation in online social interactions [32].

Research on avatars has indicated that individuals prefer personalized avatars that resemble them to represent themselves. Vasalou and Joinson discovered that users serve as the primary source of inspiration for personalized avatars [11]. As a result, users consider their avatars to be similar to themselves, despite the fact that the avatar development criteria might vary. Ratan and Dawson [24] have pointed out that users interact with self-similar

avatars on both the psychological and the physical level and that users identify themselves via the process of avatar development. Vasalou et al. showed that when participants used self-similar avatars, performance on individual and team tasks increased, and interaction with avatars increased [33]. Experiments have demonstrated that avatar similarity can improve virtual team performance and increase positive social connections among team members. Avatars with a high degree of self-resemblance may result in more concentrated attention, which may increase favorable attitudes toward virtual actions. [34]. Another experiment has also shown that the use of avatars with high self-similarity in sports can better stimulate physical activity [35].

Multiplayer online social games (MOSG) and virtual meetings for academic and professional purposes (VM) are the two virtual contexts for our experiments, i.e., a relaxed and serious social environment, and the self-similarity of avatars created in these two virtual contexts may differ. Hence, we hypothesize:

**H1a.** *Customized avatars in the context of MOSG have lower self-similarity.*

**H1b.** *Customized avatars in the context of VM have higher self-similarity.*

### 2.3. The Effect of Public Self-Consciousness on Avatar Self-Similarity

Avatars are representations of people in virtual environments and may impact how other users view them [36]. Markus and Nurius were the ones who initially presented the idea of the potential self. People are impacted by social roles and signals, and they have a desire to express themselves; their behavior is affected by the circumstances in which they find themselves. They define possible selves as a kind of self-knowledge that is concerned with how a person views their own potential and the future [37]. They offer conceptual linkages between cognition and motivation, incentives for future conduct, and spaces for individuals to analyze their present self-perceptions. Possible selves are formed using people's previous experiences and their envisioned futures [37].

The capacity to shift one's attention from the environment to oneself and back again is an essential component of self-awareness [38]. People's attention is constantly drawn inward or outward, and public and private self-consciousness is a process of self-focus [39]. The idea of self-consciousness may be separated into two categories, private and public, according to current study results. A person's private thoughts, feelings, and recollections are all instances of components of the self that are often concealed from others who are unfamiliar with the individual. That is the definition of "private self" [40,41]. Paying attention to inner thoughts and sensations is private self-consciousness. The phrase "I care passionately about the way I show myself" is an example of the public self-consciousness factor. This component is described as the general awareness of the self as a social object that has an effect on others [39]. Exposure to self-similar avatars, which may be made to resemble users, may have the same effects as mirrors and may promote public self-awareness. This is due to the fact that avatars may be created to resemble users [33]. Hence, we hypothesize:

**H2a.** *Avatar self-similarity has a negative impact on self-consciousness in the context of MOSG.*

**H2b.** *Avatar self-similarity has a positive effect on self-consciousness in the context of VM.*

### 2.4. Self-Expression

Studies have found that individuals report greater self-disclosure in the presence of less realistic avatars, feel like they are not speaking with real-life people, and experience less social anxiety and appraisal anxiety in interpersonal encounters [13]. Self-disclosure may be affected by the kind of interactive environment in which an avatar chooses to engage. In addition, research has shown that individuals who role-play using their avatars, in particular, prefer to interact in written form rather than via voice. This is completed so

that they may conceal hints about their real identities. The boundary that separates the real world from the virtual world is sometimes referred to in a metaphorical sense as a "magic circle". By using the safety that the magic circle provides, role players are able to become deeper engaged in their roles without having any effect on their actual age, gender, or inner sentiments [40]. During the COVID-19 pandemic, people were disclosing themselves on social media in order to maintain social connections with others [42].

Goffman [41] asserts that self-presentation is a theatrical or performative metaphor in which social roles perform a role in interpersonal interactions. People have to adapt their behavior to fit the context of the settings they find themselves in on a daily basis, which requires them to be aware of their appearance and to take cues from their environment. People deliberately control their appearance to maximize their capacity to attain social objectives. When avatars are used for self-presentation in a digital context, it influences how individuals choose or customize avatars [2]. Two aspects of self-expression are self-disclosure and self-presentation. In various virtual environments, users' avatars reflect varying degrees of self-expression awareness. Some people will choose avatars that correctly reflect one element of themselves but falsely represent another. Occasionally, this may be an option, but occasionally it is not. Some digital contexts make it harder to express a person's true identity due to social conventions [43]. Hence, we hypothesize:

**H3.** *Avatar self-similarity has a stronger positive impact on self-disclosure awareness in the context of MOSG.*

**H4.** *Avatar self-similarity has a stronger positive impact on self-presentation awareness in the context of VM.*

*2.5. Emotional Expression and Avatar Identification*

Emotion is the most significant factor in the realm of human interface design. Similar to how emotion recognition occurs in ordinary life, emotional stimuli can be presented in a relevant 3D environment. For instance, the physical environment, commotion, the environment's density, and the evaluation of the situation can impact emotion recognition by distracting and attracting attention [44,45]. Emotional expression has been evaluated in several ways, including emotion and speech in written language [46,47], facial expressions [48], body posture, gestures [16], physical activity [49], in visual language, and tactile perception in virtual environment [50]. Natural human communication consists of voice, facial emotions, bodily postures, and gestures. Research in cognitive psychology suggests that intrinsic internal appearance enhances human sense of homogenous species and emotion recognition [51]. Emotional valence is dynamic (moving and changing), influenced by the appearance and movement characteristics of the avatar [52].

According to the theory known as the Proteus effect, the user's behavior will adhere to the updated self-representation independent of the user's actual physical self [53,54]. Avatars of appearance [34], gender [55], race [56], or sexuality [57] alter self-perception, attitudes toward others, and behavior, according to research conducted using the Proteus effect paradigm. Avatar identification influences social behavior in virtual worlds. A higher level of avatar identification facilitates social interaction in virtual environments [33,58,59]. Through self-awareness and self-presence, the visual similarity between players and avatars facilitates their self-disclosure [14]. Individuals who create avatars that are more attractive than themselves are more sociable [60]. Avatar identification influences gamers in numerous ways, including satisfaction, loyalty, motivation, and playtime [6,9,59,61,62].

Prior research has uncovered three unique avatar identification strategies: similarity identification, embodiment identification, and wishful identification. Similarity, identification refers to the extent to which gamers think that their avatars are modeled like them in some way. Embodied identification refers to the degree to which a player feels that they are inhabiting the role of a character they are playing in a video game. Wishful identification refers to the degree to which a player's in-game avatar resembles their idealized version of

themselves. Therefore, the degree to which these identities correspond to one another is directly proportional to the degree of overlap that exists between one's actual and ideal selves, and between their bodily and emotional experiences [63]. Based on the above theory, we hypothesize:

**H5.** *Self-disclosure awareness has a stronger positive impact on avatar wishful identification in the context of MOSG.*

**H6.** *Self-presentation awareness has a stronger positive impact on avatar similarity identification in the context of VM.*

Users often use avatars to represent their ideal selves [22,60] while maintaining the aspects of their real identities that are most important to them [22]. The position, contour, shape, and eyebrows of facial features tend to have a negative correlation with avatar identification since it is difficult to modify these traits in real life. As a result, facial feature-related design elements tend to have a negative correlation with avatar identification. On the other hand, each identity was shown to have a positive correlation with hair parts that could be readily changed in the actual world [15]. Hence, we hypothesize:

**H7a.** *When customizing avatars in both virtual environments, those design elements that could easily be changed in real life were manipulated for longer periods of time.*

**H7b.** *When customizing avatars in both virtual environments, those design elements that could easily be changed in real life were more important.*

*2.6. Summary*

Figure 1 depicts the research framework and underlying hypotheses. The majority of studies hypothesize that H1–H3 and H5–H6 represent novel perspectives in the scientific literature. Despite the fact that H4 and H7a-H7b have been the subject of prior research, they are necessary for the purpose of this study. This framework contributes to the field of knowledge on avatar customization by comparing the self-similarity of avatar customization outcomes in two different virtual contexts and the influence of virtual contexts on avatar customization behaviors.

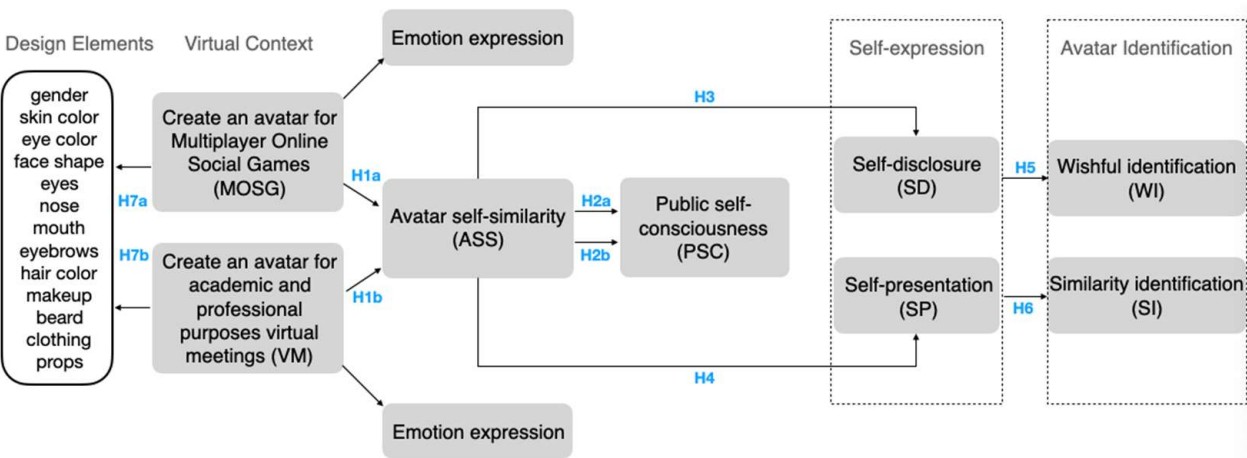

**Figure 1.** Research framework.

## 3. Materials and Methods

*3.1. Experimental Design*

This study used a 2 (gender) × 2 (context) between-subjects experimental design. One hundred undergraduate students and six graduate students (M = 35, F = 71) participated in this experiment. In order to reduce the individual differences between the subjects, when

recruiting the subjects, we asked the participants to have a certain experience of multiplayer online social games and virtual meetings and asked when their most recent experience was. Most of the participants had recent experience, as they had to use virtual meets for course learning and online gaming platforms for entertainment and social activities during the recurring COVID-19 pandemic. Participants were randomly assigned to two groups with equal proportions of males and females. We were intrigued as to whether the self-similarity of customized avatars in various virtual situations for people of different genders varied, so we also included gender as a variable. The two groups were assigned different experimental tasks. In order to reduce interference, the experimental task was administered to the two groups in separate rooms.

Before beginning the experiment, participants were asked to view a brief instructional video of the task. The video includes the procedure for utilizing the avatar customization platform and the requirements for experimental tasks. Group 1 (N = 52, M = 17, F = 35) was asked to create an avatar of their own on a 3D avatar creation platform for a specific context, i.e., for a multiplayer online social game (MOSG). Group 2 (N = 54, M = 18, F = 36) was asked to create an avatar of their own on the same 3D avatar creation platform for a specific context, i.e., virtual meetings for academic and professional purposes (VM).

Participants in both experimental groups were required to carefully experience and perceive the experimental environment. In the description video, we mentioned, "Imagine what kind of avatar you would use in such a virtual environment". Participants were told that the length of time they spent manipulating each design element would be recorded while customizing the avatar. After completing the avatar customization, participants were asked to fill out a questionnaire. After signing the informed consent form, watching the introductory video, and learning how the avatar customization platform worked, people in both groups were asked to finish customizing their avatars within 15 min.

### 3.2. Avatar Custom Platform

For this study, "Ready Player Me" served as the experimental platform. Ready Player Me is a technology company that specializes in creating customizable avatars for use in virtual reality (VR), augmented reality (AR), and other digital platforms. The company was founded by Filip Grzegorzek, Mateusz Chmiel, and Kamil Nawratil in 2017. The company's headquarters are located in Warsaw, Poland. As the metaverse is constructed, digital identities may become a crucial component, allowing users to represent themselves in virtual spaces with greater flexibility. With Ready Player Me, anyone can generate a personalized 3D avatar in mere seconds by taking a selfie or selecting from a variety of features, including physical characteristics, clothing, accessories, and NFTs owned. After creating an avatar in Ready Player Me, users can download a GLB file that can be used on multiple AR/VR platforms to interact with applications, such as online multiplayer games, social networking, and online meetings.

Reasons for selecting the Ready Player Me platform: First, the custom avatar is simple and straightforward to use. Participants can take a photo to generate an initial avatar or select from a variety of initial avatars before clicking to add design elements. Customizing an avatar is possible on computers and other mobile devices. Second, there are extensive customization options for the avatar. Figure 2 depicts the interface for configuring the avatar's skin color, eye color, face shape, facial features, hairstyle, makeup, clothing, and accessories. These are essential components for assessing self-similarity. Third, the avatar style in this instance is neutral; that is, it falls somewhere between hyperrealism and stylization. This avoids the uncanny valley effect and concerns regarding stylistic preferences [34].

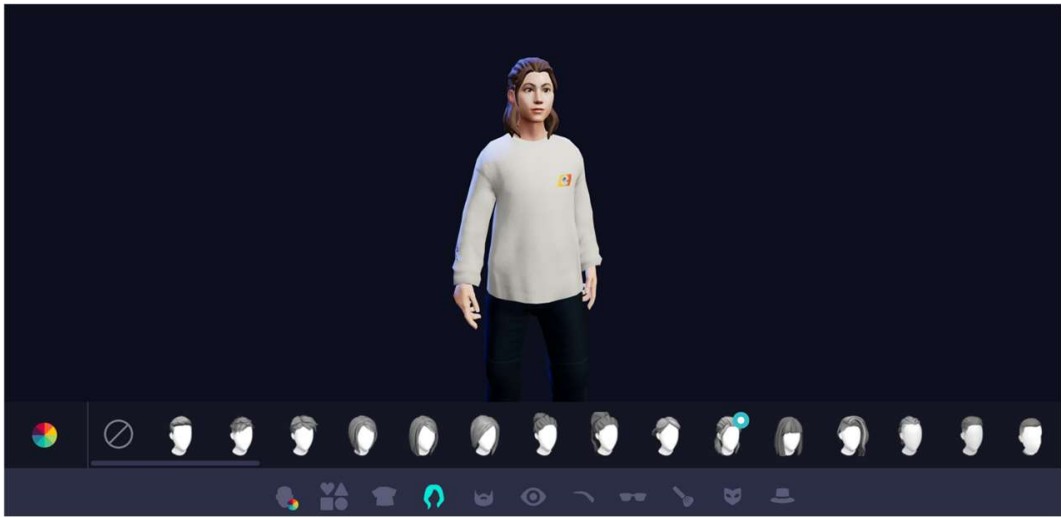

**Figure 2.** Screenshots of the avatar customization process in Ready Player Me.

### 3.3. Measurements

### 3.3.1. Self-Similarity Measurement

Avatar self-similarity was assessed on a standard scale under eight situations. We referenced and modified the experimental scale developed by Rahill and Sebrechts [7]. The features that were selected were picked because of the completeness and simplicity with which they allowed for comparisons of similarities and differences between avatars and participants. The maximum potential similarity score is 15 (15 = perfect match). Research has demonstrated that individuals utilize gender and skin color as two significant components of their avatar identification [64]. As a result, gender and skin color are given the most weight in the system, which amounts to 3 points each. Eye color, face shape, and facial characteristics are the factors with the second greatest weighting, each receiving 2 points, respectively. The variables with the lowest weighting are hair color, cosmetics, and clothes, each receiving 1 point, respectively.

### 3.3.2. Avatar Design Elements Measurement

After the experiment, participants responded to two questions in a questionnaire based on the experiment's data. They concerned the significance of avatar customization elements and the time required to select each element when customizing an avatar. On a 5-point Likert scale, the questions address every design element of the experiment.

### 3.3.3. Perception Factors Measurements

For all measures of avatar customization perception factors, variables were evaluated using a modified version of a previously validated multi-item scale. Changes have been made to the language to ensure that they are suitable for virtual environments. Throughout the procedure, the five-point Likert Scale was utilized.

We used three of the items on the Fenigstein [39] Public Self-Consciousness Scale to measure participants' public self-consciousness while customizing their avatars in the assigned virtual environment. We added a short sentence expressing the status to each item, such as "When I customize my avatar for MOSG/VM, I usually aware of my appearance". Public self-consciousness seems to be of interest to us since it is believed to be related to avatar self-similarity.

The questions we used to measure self-expression were referenced and adapted from Hooi and Cho [14] and Kim [65]. There are two variables in this question, self-disclosure and self-presentation. We add keywords to each question, such as "I want to use this avatar…", "This avatar shows…" Languages like this can help subjects recall how they felt during the process of avatar customization just now.

Similarity identification and Wishful identification were the two forms of avatar identification that we tested. In order to generate questionnaire items for each avatar identification, the first three-factor loaders from the original scale were extracted and used [63]. The phrase "while playing the game" has been replaced with "while costuming the avatar" in the question.

### 3.3.4. Emotion Expression Measurements

To investigate subjects' emotions when customizing their avatars, we used Izard's defined emotional states for emotion classification: "anger", "disgust", "fear", "guilt", "interest", "joy", "sadness" ("distress"), "shame", and "surprise" [65]. In the questionnaire, subjects were asked to choose at least one emotion expressed by the customized avatar.

## 4. Results Analysis

### 4.1. Content Reliability

After the experiment, we conducted a reliability analysis on the questionnaire data. The Cronbach alpha reliability coefficient is the most commonly used reliability coefficient. The $\alpha$ coefficient evaluates the consistency between the scores of each item on the scale and belongs to the internal consistency coefficient. This method is suitable for the reliability analysis of attitude and opinion questionnaires (scales). The reliability coefficient of the scale was preferably above 0.8, and acceptable between 0.7 and 0.8; if the Cronbach $\alpha$ coefficient was below 0.6, the questionnaire should be re-edited. According to Table 1, the Cronbach's alpha coefficients of the article's primary research variables are 0.837, 0.763, 0.826, 0.812, and 0.797, which are all greater than 0.8, indicating that the scale has good internal consistency and high reliability, and that the data could be used for further analysis.

**Table 1.** Cronbach reliability analysis.

| Variable | Number of Items | Cronbach $\alpha$ |
|---|---|---|
| Public self-consciousness | 3 | 0.837 |
| Self-disclosure | 3 | 0.763 |
| Self-presentation | 3 | 0.826 |
| Wishful identification | 3 | 0.812 |
| Similarity identification | 3 | 0.797 |

### 4.2. Content Validity

For validity analysis, we developed a confirmatory factor analysis (CFA) model (see Figure 3) based on the questionnaire content. The fundamental concept of factor analysis is to group variables based on the correlation between them so that variables in the same group have a high correlation, variables in different groups have a low correlation, and variables in each group constitute a common factor. As depicted in Figure 3, the validation factor analysis model fit index CMIN/DF = 1.48; RMSEA = 0.068; CFI = 0.942 indicates that the overall model fit is satisfactory and that the model fit has been successful.

According to Table 2, the factor loading coefficients of the obvious variables in the model are all greater than 0.5, and their significant p-values are all less than 0.05, indicating a significant relationship between the obvious and latent variables. These manifest variables can provide an explanation for their corresponding latent variables (see Table A1). Concurrently, the average variance extraction value (AVE) of each latent variable is greater than 0.5, and the combined reliability (CR) value is greater than 0.7, indicating that the scale has excellent convergent validity.

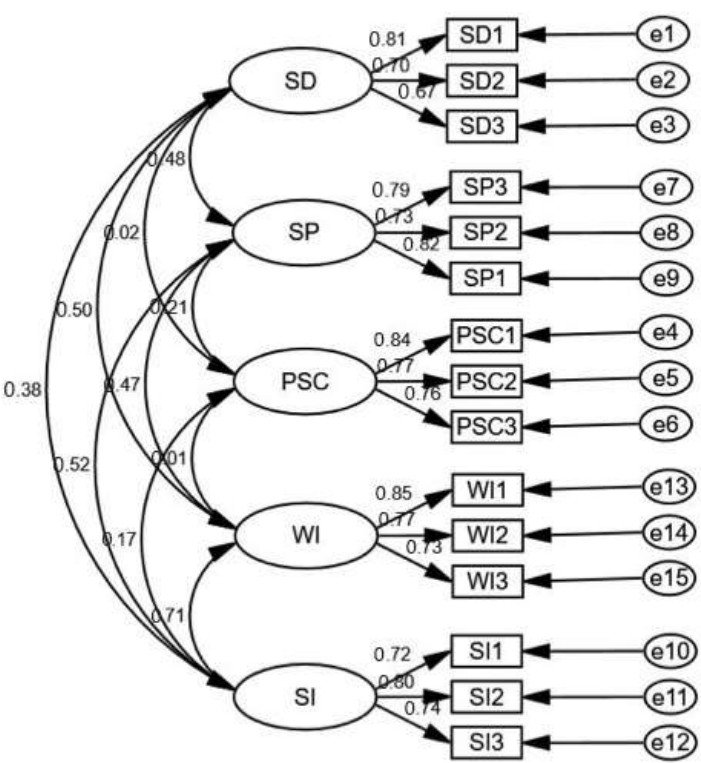

CMIN=111.134（p=0.012）; DF=80
CMIN/DF=1.389
GFI=0.886; AGFI=0.830
IFI=0.953; TLI=0.937; CFI=0.952
RMSEA=0.061

**Figure 3.** Confirmatory factor analysis model.

**Table 2.** Factor Loading Factors.

| Latent Variable | Explicit Variable | Coef. | SE | z | p | Factor Loading | AVE | CR |
|---|---|---|---|---|---|---|---|---|
| PSC | PSC1 | 1 | - | - | - | 0.845 | 0.633 | 0.838 |
| | PSC2 | 0.936 | 0.122 | 7.699 | 0 | 0.775 | | |
| | PSC3 | 0.928 | 0.121 | 7.638 | 0 | 0.765 | | |
| SD | SD1 | 1 | - | - | - | 0.811 | 0.533 | 0.773 |
| | SD2 | 0.988 | 0.161 | 6.137 | 0 | 0.697 | | |
| | SD3 | 0.914 | 0.152 | 6.006 | 0 | 0.674 | | |
| SP | SP1 | 1 | - | - | - | 0.818 | 0.612 | 0.825 |
| | SP2 | 0.94 | 0.127 | 7.373 | 0 | 0.735 | | |
| | SP3 | 0.986 | 0.126 | 7.823 | 0 | 0.792 | | |
| WI | WI1 | 1 | - | - | - | 0.846 | 0.614 | 0.826 |
| | WI2 | 1.136 | 0.14 | 8.111 | 0 | 0.765 | | |
| | WI3 | 1.15 | 0.148 | 7.778 | 0 | 0.735 | | |
| SI | SI1 | 1 | - | - | - | 0.718 | 0.569 | 0.798 |
| | SI2 | 1.301 | 0.186 | 6.997 | 0 | 0.803 | | |
| | SI3 | 1.197 | 0.18 | 6.634 | 0 | 0.739 | | |

Note. PSC = public self-consciousness; SD = self-disclosure; SP = self-presentation; SI = similarity identification; WI = wishful identification.

According to Table 3, the correlation coefficients of the principal variables examined in this study were all less than the square root of their respective AVE, indicating that the discriminant validity between the latent variables is good.

**Table 3.** Discriminant validity: Pearson correlation and AVE square root value.

|  | Factor 1 | Factor 2 | Factor 3 | Factor 4 | Factor 5 |
|---|---|---|---|---|---|
| PSC | 0.795 |  |  |  |  |
| SD | 0.005 | 0.724 |  |  |  |
| SP | 0.174 | 0.387 | 0.781 |  |  |
| WI | 0.02 | 0.413 | 0.372 | 0.773 |  |
| SI | 0.128 | 0.286 | 0.41 | 0.573 | 0.758 |

Note. PSC = public self-consciousness; SD = self-disclosure; SP = self-presentation; SI = similarity identification; WI = wishful identification. The bold number on the diagonal represents the AVE square root value.

### 4.3. Avatar Self-Similarity

Table 4 displays the results of testing the moderating effect of gender on the relationship between context and self-similarity using 2 (gender) × 2 (context) two-factor variances. The impact of context and gender on self-similarity was examined using a two-way ANOVA. Table 4 shows that context is significant (F = 31.612, $p < 0.05$). It demonstrates the main effect's existence and the context's differential relationship with self-similarity. There is no gender distinction (F = 2.272, $p > 0.05$). It demonstrates that self-similarity and gender have no differential relationship. The context is significant, as is its interaction with gender (F = 4.922, $p < 0.05$). It shows that gender and context have significant second-order effects on self-similarity.

**Table 4.** 2 (gender) × 2 (context) Two-way ANOVA analysis results.

| Source of Difference | Sum of Square | df | Mean Square | F | p |
|---|---|---|---|---|---|
| Intercept | 5158.512 | 1 | 5158.512 | 488.988 | 0.000 *** |
| gender | 23.964 | 1 | 23.964 | 2.272 | 0.135 |
| context | 333.486 | 1 | 333.486 | 31.612 | 0.000 *** |
| gender × context | 51.921 | 1 | 51.921 | 4.922 | 0.029 * |
| Residual | 1076.034 | 102 | 10.549 |  |  |

$R^2$: 0.255. Note. * $p < 0.05$ *** $p < 0.001$.

According to Table 5, the self-similarity of male-customized avatars in the MOSG context (5.29) differs significantly from that in the VM context (10.56), and the self-similarity of the VM avatar will be significantly greater than that of the MOSG avatar. There is a significant difference in self-similarity between the MOSG context (5.77) and the VM context (5.00) for female-customized avatars (8.06). Overall, the mean difference in self-similarity between avatars customized by men in the two contexts is −5.261, which is significantly greater than the mean difference between female-customized avatars, which is −2.284. In conclusion, the H1a and H1b are supported.

**Table 5.** Mean Contrast by Gender and Context (Mean ± SD).

|  | MOSG (n = 52) | VM (n = 54) | Mean Difference | SE | t | p |
|---|---|---|---|---|---|---|
| Male | 5.29 ± 1.69 | 10.56 ± 5.00 | −5.261 | 1.098 | −4.79 | 0 |
| Female | 5.77 ± 2.22 | 8.06 ± 3.54 | −2.284 | 0.771 | −2.963 | 0.004 |

Note. MOSG = Multiplayer Online Social Gaming; VM = Work-study Virtual Meeting.

### 4.4. Structural Equation Modeling

To determine the effects of avatar self-similarity on self-consciousness, self-disclosure, and self-presentation, and whether the effects of self-presentation on avatar identification vary by virtual context, this study used AMOS 23.0 software to construct a structural equation model, using a multi-group A group structural equation model was used to compare the two experimental groups. The structural equation model of the MOSG

context experiment is shown in Figure 4. The structural equation model of the VM context experiment is shown in Figure 5.

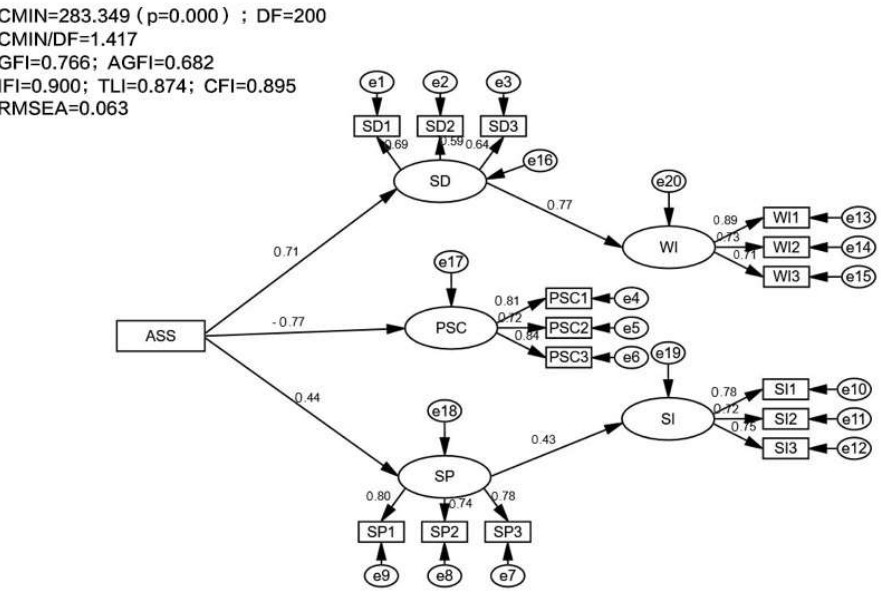

**Figure 4.** The structural equation model of the MOSG context experiment.

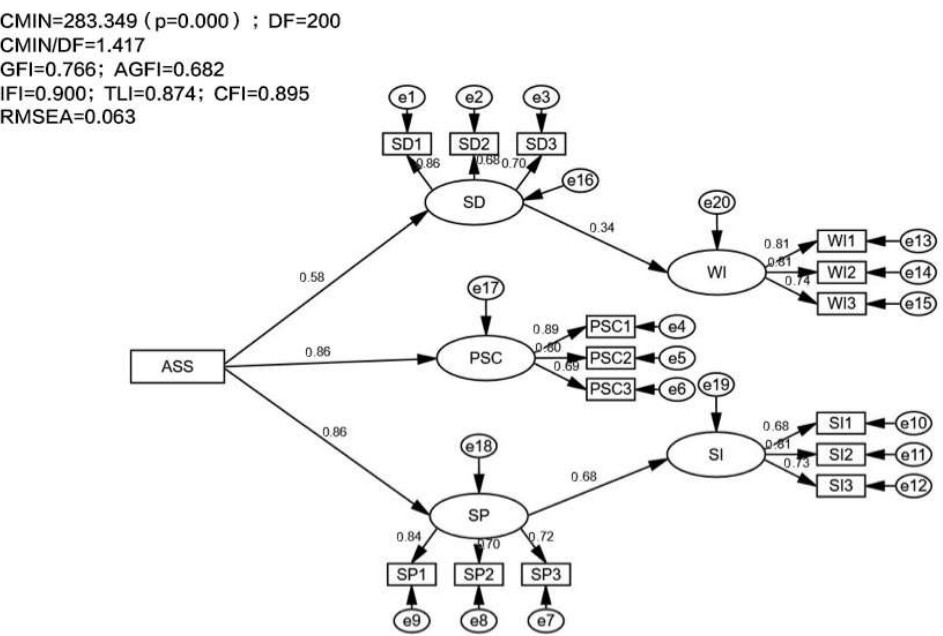

**Figure 5.** The structural equation model of the VM context experiment.

### 4.4.1. The Effect of Avatar Self-Similarity on Self-Consciousness

As shown in Table 6, when avatars were customized in the MOSG context, the standardized path coefficient of ASS on PSC was −0.768, indicating that ASS has a statistically significant negative effect on PSC ($p < 0.05$); therefore, H2a is supported. The standardized path coefficient of ASS on PSC in the VM context is 0.864, indicating that ASS had a significant positive effect on PSC ($p < 0.05$); thus, H2b was supported.

**Table 6.** Comparison of standardized path parameter estimation and T values for multigroup structural equation models.

| Path | | | MOSG | | VM | |
|---|---|---|---|---|---|---|
| | | | Path Coefficient β | T Statistics | Path Coefficient β | T Statistics |
| ASS | → | SD | 0.712 | 4.335 | 0.579 | 4.313 |
| ASS | → | SP | 0.442 | 2.955 | 0.865 | 5.939 |
| ASS | → | PSC | −0.768 | −5.923 | 0.864 | 8.753 |
| SP | → | SI | 0.427 | 2.41 | 0.676 | 3.464 |
| SD | → | WI | 0.769 | 4.016 | 0.339 | 2.043 |

Note. ASS = Avatar self-similarity; PSC = public self-consciousness; SD = self-disclosure; SP = self-presentation; SI = similarity identification; WI = wishful identification; MOSG = Multiplayer Online Social Gaming; VM = Work-study Virtual Meeting.

#### 4.4.2. The Effect of Avatar Self-Similarity on Self-Expression

In the MOSG context, the standardized path coefficient of ASS on SD was 0.712, indicating that ASS had a statistically significant positive effect on SD ($p < 0.05$). In the VM context, the standardized path coefficient of the positive effect of ASS on SD was 0.579, indicating that ASS had a significant positive effect on SD ($p < 0.05$). Comparing the two path coefficients (0.712 > 0.579), the positive relationship between ASS and SD was stronger when customizing the avatar in the MOSG context. When customizing avatars for VMs, the positive correlation between ASS and SD was weakened. Therefore, it is presumed that H3 is supported.

The standardized path coefficient of ASS on SP in the MOSG context was 0.442, indicating that ASS had a significant positive effect on SP ($p < 0.05$). The standardized path coefficient of ASS on SP in the VM context was 0.865, indicating that ASS had a significant positive effect on SP ($p < 0.05$). According to the path coefficient (0.442 < 0.865), which indicates that the positive effect of ASS on SP was weaker when the avatar is customized in the MOSG context. The positive effect of ASS on SP was stronger when avatars were customized in the VM context. Hence, H4 was supported.

#### 4.4.3. The Effect of Self-Expression on Avatar Identification

The standardized path coefficient of SD on WI in the MOSG context was 0.769, indicating that SD had a significant positive effect on WI ($p < 0.05$). In the VM context, the standardized path coefficient of SD on WI was 0.339, indicating that SD significantly increased WI ($p < 0.05$). Comparing the two path coefficients revealed that SD had a greater positive effect on WI when avatars were customized within the MOSG context (0.769 > 0.339). When avatars were customized in the context of VM, the positive relationship between SD and WI was weakened. The results showed that H5 was supported.

The standardized path coefficient of SP on SI in the MOSG context was 0.427, indicating that SP had a significant positive effect on SI ($p < 0.05$). In the context of VM, the standardized path coefficient of SP on SI was 0.676, indicating that SP had a statistically significant positive effect on SI ($p < 0.05$). Comparing the two path coefficients (0.427 < 0.676) revealed that the positive relationship between SP and SI was weak when the MOSG avatars were customized. When avatars were customized in the context of VM, the positive relationship between SP and SI was strengthened. H6 was ultimately supported.

#### 4.5. Independent Sample t-Test

#### 4.5.1. *t*-Test for Avatar Design Element Importance

As shown in Table 7, the *t*-test was used to investigate the differences in the importance of 13 avatar design elements in the experiment in the two contexts. The samples in two contexts, nine design elements did not show significant ($p > 0.05$). Gender, skin color, eye color, face shape, eyes, nose, mouth, eyebrows, and beard all showed consistency and no differences. Four items, hairstyle, makeup, clothing, and props, showed significance

($p < 0.05$). Hairstyle showed significant at 0.01 level (t = 7.524, $p$ = 0.000 ***). The makeup effect showed a significance level of 0.01 (t = 7.420, $p$ = 0.000 ***). Clothing showed significance at the 0.05 level (t = 2.287, $p$ = 0.024 *). Prop showed significance at 0.01 level (t = 4.226, $p$ = 0.000 ***). In the MOSG context, hairstyle (4.37), makeup (4.13), and clothing (4.58) have higher values than other elements. In the VM context, clothing (4.19), gender (3.67) and skin color (3.67) have higher values than other elements. It shows that these elements are more important in the corresponding experimental context. The results showed that H7b was not supported.

**Table 7.** The *t*-test analysis results of the importance of avatar design elements.

| | Context (Mean ± Standard Deviation) | | t | p |
| --- | --- | --- | --- | --- |
| | MOSG (*n* = 52) | VM (*n* = 54) | | |
| Gender | 3.73 ± 0.95 | 3.67 ± 1.29 | 0.292 | 0.771 |
| Skin color | 3.73 ± 0.82 | 3.67 ± 1.06 | 0.348 | 0.728 |
| Eye color | 3.65 ± 1.05 | 3.44 ± 1.08 | 1.016 | 0.312 |
| Face shape | 3.58 ± 0.89 | 3.44 ± 0.92 | 0.750 | 0.455 |
| Eyes | 3.31 ± 0.96 | 3.26 ± 0.89 | 0.269 | 0.789 |
| Nose | 3.27 ± 0.91 | 3.26 ± 0.89 | 0.057 | 0.955 |
| mouth | 3.12 ± 1.02 | 3.30 ± 0.94 | −0.947 | 0.346 |
| Eyebrows | 3.27 ± 1.03 | 3.41 ± 1.07 | −0.676 | 0.501 |
| Hairstyle | 4.37 ± 0.82 | 2.61 ± 1.50 | 7.524 | 0.000 *** |
| Makeup | 4.13 ± 0.79 | 2.48 ± 1.42 | 7.420 | 0.000 *** |
| Beard | 2.62 ± 1.16 | 2.96 ± 1.27 | −1.469 | 0.145 |
| Clothing | 4.58 ± 0.70 | 4.19 ± 1.03 | 2.287 | 0.024 * |
| Props | 3.62 ± 1.29 | 2.59 ± 1.21 | 4.226 | 0.000 *** |

Note. * $p < 0.05$ *** $p < 0.001$.

### 4.5.2. *t*-Test for Manipulation Duration of Avatar Design Elements

According to Table 8, the eyes, nose, mouth, eyebrows, and beard of the two context samples were not statistically different ($p > 0.05$), indicating that there was no difference between the two experimental groups. Eight items in the context sample showed significant ($p < 0.05$). Gender was significant at 0.01 level (t = 2.950, $p$ = 0.004 **). Skin color showed significance at 0.01 level (t = 4.162, $p$ = 0.000 ***). Eye color showed significance at 0.01 level (t = 4.994, $p$ = 0.000 ***). Face shape showed significance at 0.01 level (t = 2.902, $p$ = 0.005 **). Hair color showed a 0.01 level significance (t = 6.509, $p$ = 0.000 ***). Makeup effect showed a significance level of 0.01 (t = 10.611, $p$ = 0.000 ***). Clothing showed significance at 0.05 level (t = 2.566, $p$ = 0.012 *). Props showed significance at 0.01 level (t = 5.808, $p$ = 0.000 ***). In the context of the MOSG, hairstyle (4.31), makeup (4.50), and clothing (4.15) have greater values than other elements. In the context of the VM, hairstyle (3.04), makeup (2.63), and clothing (3.59) have greater values than other elements. It demonstrates that these elements are manipulated for an extended period of time in the corresponding experimental setting; therefore, H7a was supported.

**Table 8.** The *t*-test analysis results of the manipulation duration of the avatar design elements.

| | Context (Mean ± Standard Deviation) | | t | p |
| --- | --- | --- | --- | --- |
| | MOSG (*n* = 52) | VM (*n* = 54) | | |
| Gender | 1.85 ± 0.87 | 1.37 ± 0.78 | 2.950 | 0.004 ** |
| Skin color | 2.73 ± 1.17 | 1.89 ± 0.88 | 4.162 | 0.000 *** |
| Eye color | 3.04 ± 1.20 | 2.00 ± 0.91 | 4.994 | 0.000 *** |

**Table 8.** *Cont.*

| | Context (Mean ± Standard Deviation) | | t | p |
|---|---|---|---|---|
| | **MOSG (*n* = 52)** | **VM (*n* = 54)** | | |
| Face shape | 3.08 ± 0.93 | 2.59 ± 0.79 | 2.902 | 0.005 ** |
| Eyes | 2.65 ± 0.84 | 2.52 ± 0.88 | 0.808 | 0.421 |
| Nose | 2.54 ± 0.80 | 2.41 ± 0.79 | 0.847 | 0.399 |
| Mouth | 2.50 ± 0.75 | 2.44 ± 0.88 | 0.348 | 0.729 |
| Eyebrows | 2.65 ± 0.84 | 2.52 ± 0.97 | 0.769 | 0.443 |
| Hairstyle | 4.31 ± 0.78 | 3.04 ± 1.18 | 6.509 | 0.000 *** |
| Makeup | 4.50 ± 0.75 | 2.63 ± 1.03 | 10.611 | 0.000 *** |
| Beard | 2.00 ± 1.01 | 1.96 ± 1.12 | 0.179 | 0.858 |
| Clothing | 4.15 ± 1.18 | 3.59 ± 1.07 | 2.566 | 0.012 * |
| Props | 3.50 ± 1.57 | 2.00 ± 1.03 | 5.808 | 0.000 *** |

Note. * $p < 0.05$ ** $p < 0.01$ *** $p < 0.001$.

## 5. Discussion

### 5.1. Findings and Theoretical Implications

The study provided reasonable support for most of the hypotheses, with only one not being supported. Studies have shown that avatar self-similarity can be affected by the virtual environment. In different virtual environments, avatar self-similarity has different effects on public self-awareness and self-expression. Self-expression also had different effects on avatar identification.

#### 5.1.1. Variations in Avatar Self-Similarity across Virtual Contexts

Previous studies have demonstrated that users favor individualized avatars that are based on their own self-images [11]. Users interact with self-similar avatars at both the mental and physical levels, and users identify themselves through the development of avatars [66]. This study provides some support for these viewpoints. Our experiments take place in two different settings: multiplayer online social games and work–study meetings. The findings show that avatars in both groups of experiments are self-similar. The subjects' customized avatars share similarities with themselves in a number of ways. When comparing the results of the two experimental groups, it is intriguing to observe that our hypothesis is supported by both. In the context of the MOSG experiment, the avatar's self-similarity is low (Figure 6). Many subjects chose a different color than their own skin, eyes, and even a different gender. In terms of hairstyle, makeup, and clothing, most of the subjects chose to use things that were not available or not common in real life. Within the context of the virtual reality experiment, avatar self-similarity is high (Figure 7). The subjects essentially chose the same skin tone, eye color, and gender as themselves. The subjects chose to use some common or similar design elements in their avatar shape and clothing. We also compared the self-similarity of male and female avatars in the two experiments out of curiosity. Although males and females differed in avatar self-similarity in both experiments, males had greater differences than females in both experiments.

#### 5.1.2. The Effect of Avatar Self-Similarity on Self-Consciousness and Self-Expression

Since avatars may be designed to resemble users, exposure to self-similar avatars may have the same effects as mirrors and may increase public self-consciousness [33]. This study validates these views. In the context of the VM experiment, the subject's customized avatar was tested with higher self-similarity. In this context, subjects may perceive this as a more formal and serious environment than a social gaming environment. We also tested that the experimenter's perception of public self-consciousness was higher in this experimental context, so our hypothesis was supported. Subjects need to customize avatars that are more similar to themselves in real life in order to better present their social roles and show a higher awareness of self-presentation. Due to such a context, the subjects' public self-consciousness is heightened, and they do not want to disclose their private side or

information that might be criticized by those they know. Therefore, the customized avatar Is visually similar to the self in real life, showing a higher self-presentation awareness.

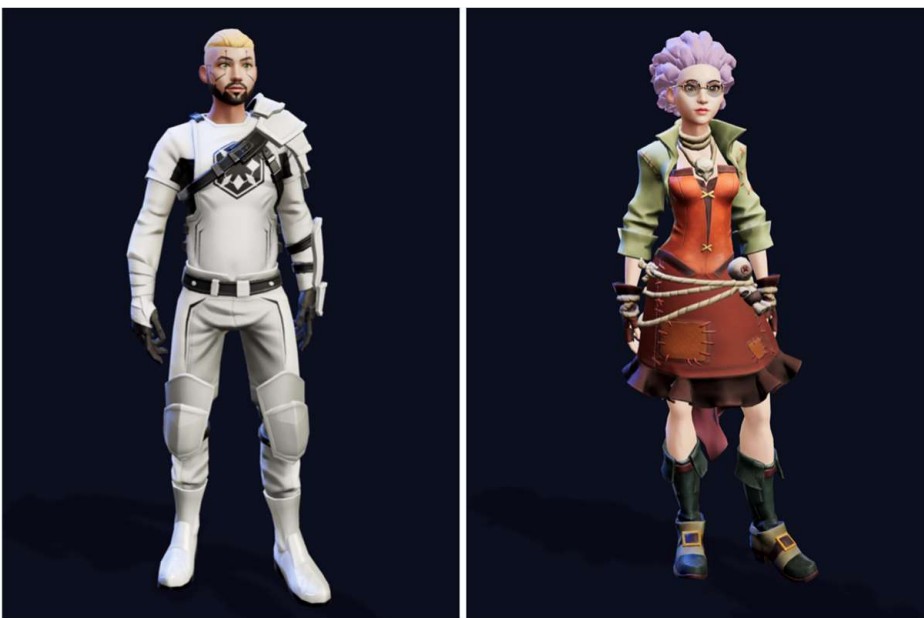

**Figure 6.** Avatar samples in the context of the MOSG experiment.

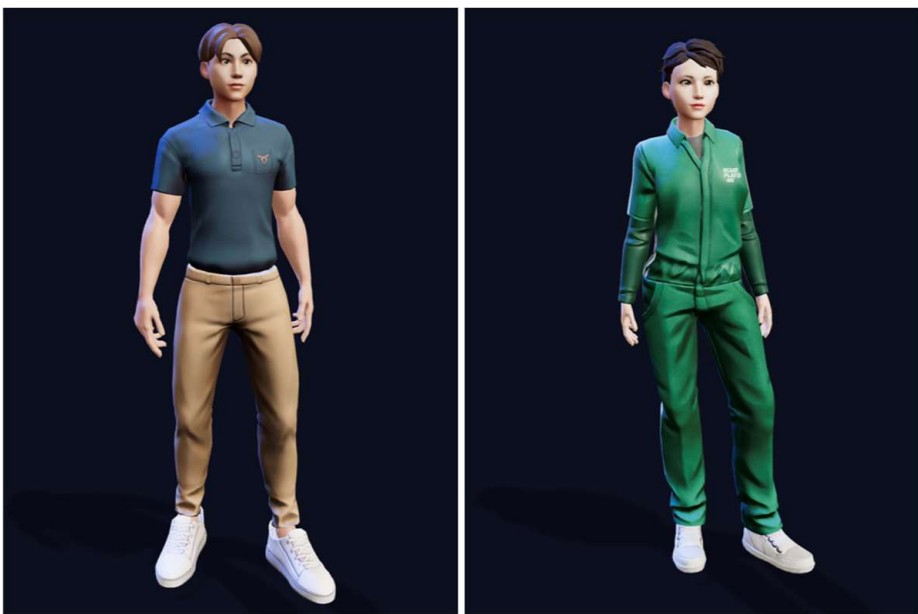

**Figure 7.** Avatar samples in the context of the VM experiment.

On the contrary, in the context of the MOSG experiment, the environment perceived by the subjects is relatively relaxed and free, and the subjects' public self-consciousness of the test source shows a lower level. Research has shown that it has become easier to pass sensitive personal data anonymously on the Internet without fear of punishment or reprimand [67]. The anonymous communication nature of the Internet has led to an increased awareness of self-disclosure. As in the context of our MOSG experiment, when the subjects perceive that this will be an environment with no acquaintances and friends and can use this avatar to disclose themselves or role-play in an unfamiliar environment, the customized avatar will be more idealized. Therefore, avatars with low self-similarity have a stronger effect on self-disclosure.

### 5.1.3. The Effect of Self-Expression on Avatar Identification

Similarity identification is how similar gamers perceive their avatars to be. Wishful identification is how closely a game's in-game avatar resembles their ideal self. The degree to which these identities correspond is proportional to the overlap between a person's actual and ideal selves and physical and emotional experiences [63]. In this study, we found that in the context of the MOSG experiment, because the subjects had a strong willingness to self-disclosure, they used their own image as inspiration when customizing their avatars and, at the same time, extended to a more ideal design of themselves, which is more effective express their true inner image. Therefore, the subjects had a higher identification with the avatar ideal. In contrast, in the context of the VM experiment, the subjects showed a higher awareness of self-presentation and higher identification with avatar similarity.

### 5.1.4. Avatar Customization and Emotion Expression

Previous research has shown that design elements that are easily changed in real life are positively correlated with all identities [15]. Our experiments verify this. In our two sets of experiments, the three elements of hairstyle, clothing, and makeup were manipulated for a longer time, which is in line with our hypothesis. Our other hypothesis, that these three elements show high importance in both sets of experiments, was not supported. The results show that only in the context of the MOSG experiment, these three elements present a high level of importance, but in the context of the VM experiment, gender, skin color, and clothing are more important. According to other studies, in the context of the VM experiment, the self-similarity of the avatar customized by the subjects is higher, and gender and skin color are also the two highest scores in the self-similarity measurement. Therefore, the similarity of gender and skin color determines the high self-similarity of avatars.

In the experiment, we observed that in the MOSG experimental context, some subjects showed higher motivation and more active communication, but some subjects showed more need for privacy to complete the experiment. These are related to the personality and psychology of the subjects. However, we also observed that in this group of experiments, almost all the subjects completed in the last 1 or 2 min of the prescribed experimental time. This shows that in order to better present themselves, they need more time to think and know themselves. Our results also compare that this group manipulated each element for longer periods of time. Compared with the VM experimental context group, we observed that most subjects were not very excited about avatar customization. Most of the subjects used photos to generate initial avatars. In this platform, the initial avatars generated by AI-recognized portraits are very similar, then they focus on adjusting hairstyles, clothing, and makeup, and it takes a short time to complete the experiment.

In terms of emotional expression, we were pleasantly surprised to find that most of the subjects in the two groups of experiments chose positive emotional words in the list, such as "happiness", "surprise", and "interest". In the VM experimental group, very few subjects chose "fear" and "shame". This may be due to the perception of virtual context and the personality of the subjects. Combined with other research, we found that although the expression of emotions can be influenced by the virtual environment, the visual appeal, variety, and flexibility of avatar customization elements may prompt users to express positive emotions. In the future, there will be a variety of virtual identities, and virtual avatars will represent us in various virtual places. Avatar customization will be an area that designers and developers are constantly exploring and practicing. This research will provide some contributions to avatar customization.

### 5.2. Research Limitations

The findings of the study are favorable, but there are limitations. Due to time and financial constraints, only 106 students participated in this experiment. If a larger experimental population is included, the perception of virtual environments by individuals of various ages and occupations may be more diverse. This investigation was restricted to avatar customization. Future research can investigate the use of avatars in different virtual

environments in greater depth. Avatars designed for the same platform are encountered in distinct virtual environments and may offer unique perspectives.

## 6. Conclusions

This research shows that users perceive an upcoming virtual environment before customizing their avatar on a third-party platform. Even if the avatars are customized on the same platform, the visual similarity between the avatar and the user will vary based on the difference in the user's perception of the virtual environment. The virtual environment will affect users' public self-consciousness. In a more relaxed and comfortable virtual social environment, users have lower public self-consciousness and higher self-disclosure awareness and are more likely to customize their ideal or fantasy avatars. The low self-similarity of these avatars reflects that each design element of custom avatars should be more diverse and less restrictive. In a more formal and serious virtual social environment, users have a higher public self-consciousness, exhibit a higher awareness of self-presentation, and are more likely to customize avatars that are similar to or can introduce themselves. These avatars have a greater degree of self-similarity and require less time to select certain design elements, such as gender, skin color, etc. Choosing to manipulate hairstyles, clothes, and other elements that can easily be changed in the real world takes more time, reflecting the importance of these elements. Although emotional expression will be affected by the environment, using avatars as an interactive medium in the virtual world, the emotions expressed by users when customizing avatars are positive, and avatars can reduce social anxiety to a certain extent. Everyone can use avatars to represent their various virtual identities in future virtual interactions. The first step for a user to experience virtual interaction is to experience avatar identification in a custom avatar. The customizability and visual effects of avatars may also affect user preferences.

**Author Contributions:** Conceptualization, S.W. and Y.P.; methodology, S.W.; software, S.W. and Z.D.; validation, S.W., formal analysis, S.W. and L.X.; investigation, S.W.; resources, S.W.; data curation, S.W. and L.X.; writing—original draft preparation, S.W.; writing—review and editing, S.W. and L.X.; visualization, S.W.; supervision, Y.P.; project administration, S.W.; funding acquisition, S.W. All authors have read and agreed to the published version of the manuscript.

**Funding:** This research was partially supported by College of Chinese and ASEAN Arts, Chengdu University, the major scientific research achievements.

**Institutional Review Board Statement:** The study was conducted in accordance with the Declaration of Helsinki, and approved by the Ethics Committee of Chengdu University.

**Informed Consent Statement:** Informed consent was obtained from all subjects involved in the study.

**Data Availability Statement:** Not applicable.

**Acknowledgments:** We thank Younghwan Pan for his guidance and assistance with the content of the research. Additionally, we would like to thank all the student volunteers who participated in the experiment.

**Conflicts of Interest:** The authors declare no conflict of interest.

## Appendix A

**Table A1.** Wording for Variables.

| Construct | Items | Wording | Source |
|---|---|---|---|
| Public self-consciousness | PSC1 | I care about what other people think of my avatar. | [39] |
| | PSC2 | I care what other people think of me. | |
| | PSC3 | I worry about others seeing my flaws. | |

**Table A1.** *Cont.*

| Construct | Items | Wording | Source |
|---|---|---|---|
| Self-disclosure | SD1<br>SD2<br>SD3 | I want to use this avatar to express my private side.<br>I want to express my true self with this avatar.<br>This avatar can show what I cannot quite show in real life. | [14] |
| Self-presentation | SP1<br>SP2<br>SP3 | This avatar represents a me in real life.<br>This avatar can introduce myself.<br>Anyone who sees this avatar will know it is me. | [65] |
| Wishful identification | WI1<br>WI2<br>WI3 | The avatar I created is my ideal self.<br>The avatars I create have the traits I would like to have.<br>This avatar is what I want to be. | [26] |
| Similarity identification | SI1<br>SI2<br>SI3 | This avatar is related to who I am in real life.<br>This avatar looks a lot like me.<br>This avatar resembles me in many ways. | [15] |

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
