# Peer review of "Factors Affecting Avatar Customization Behavior in Virtual Environments"

_electronics, doi:10.3390/electronics12102286_

Round 1
Reviewer 1 Report
The paper is very nicely written and easy to follow. It provides a clear introduction and an extensive presentation of the hypotheses and the experimental results. The authors had an adequate amount of volunteers in the study and discuss the results in a clear and easy-to-convey manner.
Weak Points
- Is the mention of "self-self-consciousness" intentional? Or is the one "self" redundant? If it is not a mistake, it should be analyzed better before hypothesis 2.
- Even though the authors provide an extensive review of the related literature, there should be more recent studies included. Most of the cited works are more than a decade old, and in the past 10 years, many studies on emotional reactivity have been published that could be referred to in this paper.
Author Response
We thank you for your comments on our manuscript "Factors Affecting Avatar Customization Behavior in Virtual Environments" (2398798).Those comments are valuable and helpful.For the reviewer's response, please see the attachment.

Reviewer 2 Report
A really good piece of work, as this is a contemporary topic please make sure you use uptodate literature after 2023. You could also use the same study for different tie periods, and countries please keep up the good work
Reviewer 3 Report
Was the subject experienced in virtual meetings with avatars and MOSG? Inexperienced subjects may not be able to imagine the scene exactly as it is supposed to be. It is recommended that these factors have little effect on analysis results, or that additional consideration be given to them.
Author Response
We thank you for your comments on our manuscript "Factors Affecting Avatar Customization Behavior in Virtual Environments" (2398798).Those comments are valuable and helpful. For the reviewer's response, please see the attachment.
Reviewer 4 Report
This study aims to analyze the factors that affect the behavior of user-customized avatars in different virtual environments.
The differences in public self-consciousness, self-expression, and emotional expression among customized avatars in multiple virtual contexts are compared.
Two random groups of participants were asked to customize avatars for themselves in two contexts, a multiplayer online social game (MOSG) and a virtual meeting (VM).
When subjects perceived a more relaxed environment, the customized avatars had less self-similarity, and the subjects exhibited a stronger self-disclosure willingness and enhanced avatar wishful identification on the other hand, public self-consciousness was not increased.
When subjects perceived a more serious environment, the customized avatars exhibited a higher degree of self-similarity, and the subjects exhibited a greater self-presentation willingness, along with enhanced identification of avatar similarity, and increased public self-consciousness.
The study provided reasonable support for most of the hypotheses. Studies have shown that avatar self-similarity can be affected by the virtual environment. In different virtual environments, avatar self-similarity has different effects on public self-awareness and self-expression. Self-expression also had different effects on avatar identification.
The structure of the paper is theoretically consistent. Theoretical background and hypotheses are well defined. The research framework and underlying hypotheses are depicted in figure 1 clearly. Experimental Design, Avatar Custom Platform, Self-similarity measurement, Perception factors measurements, Emotion expression measurements are explained well. It is interesting approach to analyze by AMOS23 software to construct a structural equation model. A group structural equation model was used to compare the two experimental groups and the results are analyzed clearly.
Suggestion is
In abstract …
“This study aims to analyze the factors that affect the behavior of user-customized avatars in different virtual environments, and compare the differences in public self- consciousness, self-expression, and emotional expression among customized avatars in multiple virtual contexts.” is a method
What is your motivation? What do you want to solve the problem or realize?
I think that there is several typo.
Author Response

(The authors gave the same response as above.)
